# Foot-and-Mouth Disease Virus 3A Hijacks Sar1 and Sec12 for ER Remodeling in a COPII-Independent Manner

**DOI:** 10.3390/v14040839

**Published:** 2022-04-18

**Authors:** Heng-Wei Lee, Yi-Fan Jiang, Hui-Wen Chang, Ivan-Chen Cheng

**Affiliations:** 1School of Veterinary Medicine, National Taiwan University, Taipei 106, Taiwan; d06629010@ntu.edu.tw (H.-W.L.); yfjiang@ntu.edu.tw (Y.-F.J.); huiwenchang@ntu.edu.tw (H.-W.C.); 2Graduate Institute of Molecular and Comparative Pathobiology, School of Veterinary Medicine, National Taiwan University, Taipei 106, Taiwan

**Keywords:** foot-and-mouth disease virus 3A protein, COPII factors, ER remodeling

## Abstract

Positive-stranded RNA viruses modify host organelles to form replication organelles (ROs) for their own replication. The enteroviral 3A protein has been demonstrated to be highly associated with the COPI pathway, in which factors operate on the ER-to-Golgi intermediate and the Golgi. However, Sar1, a COPII factor exerting coordinated action at endoplasmic reticulum (ER) exit sites rather than COPI factors, is required for the replication of foot-and-mouth disease virus (FMDV). Therefore, further understanding regarding FMDV 3A could be key to explaining the differences and to understanding FMDV’s RO formation. In this study, FMDV 3A was confirmed as a peripheral membrane protein capable of modifying the ER into vesicle-like structures, which were neither COPII vesicles nor autophagosomes. When the C-terminus of 3A was truncated, it was located at the ER without vesicular modification. This change was revealed using mGFP and APEX2 fusion constructs, and observed by fluorescence microscopy and electron tomography, respectively. For the other 3A truncation, the minimal region for modification was aa 42–92. Furthermore, we found that the remodeling was related to two COPII factors, Sar1 and Sec12; both interacted with 3A, but their binding domains on 3A were different. Finally, we hypothesized that the N-terminus of 3A would interact with Sar1, as its C-terminus simultaneously interacted with Sec12, which could possibly enhance Sar1 activation. On the ER membrane, active Sar1 interacted with regions of aa 42–59 and aa 76–92 from 3A for vesicle formation. This mechanism was distinct from the traditional COPII pathway and could be critical for FMDV RO formation.

## 1. Introduction

All positive-stranded RNA viruses, including enterovirus and foot-and-mouth disease virus (FMDV) (family *Picornaviridae*), modify the host organelle to achieve their own replication. The modified organelles are termed “replication organelles” (ROs) [1,2]. It has been postulated that ROs could serve as platforms for recruiting key components for viral replication, concentrating resources in favor of the virion package [3,4], and shielding the viral RNA from RNases and RNA sensors [5]. Much is known about the mechanism of RO formation for enteroviruses (including poliovirus (PV), coxsackie B virus (CBV), and rhinovirus). For example, the PV 3A protein has been observed to “hijack” COPI factor GBF1, resulting in the inhibition of Arf1 activation [6] and secretory blockage [7]. However, the biological function should be verified. In a recent study, it was suggested that Arf1 activation directly supported RO formation [8]. In addition, 3A accumulation required 3A–GBF1 interaction [9]. However, PV 3A also interacted with ACDB3, further recruiting PI4KB for phosphorylating phosphoinositol (PI) lipids into PI4P [10,11], which might attract viral 3D polymerase [11], 3C protease [12], and host protein OSBP [13]. OSBP further exchanged PI4P for cholesterol from the endoplasmic reticulum (ER). The cholesterol accumulation mediated the processing of the viral precursor protein [13] and could be one of the factors in RO formation. Collectively, the viral 3A (or 3AB) protein could represent an RO marker protein that is highly associated with enteroviral RO development [13,14]. However, this has not been proven in FMDV. Numerous differences between FMDV and enterovirus exist: (1) FMDV has been shown to be resistant to GBF1 inhibitors [15,16] and highly associated with the COPII pathway, rather than COPI [16]. (2) The combination of FMDV 2B and 2C, instead of 3A, can block the secretory pathway in the TsO45 VSV G protein trafficking model [17,18]. (3) According to the results of the transmission electron microscopy (TEM) from Monaghan et al., FMDV RO is homologous and gathers beside the nucleus, while enteroviral RO appears as heterologous vesicles or tubules [2,19]. (4) FMDV does not require PI4K or PI4P [20,21]. (5) The C-terminus of FMDV 3A is longer than the C-termini of other picornaviruses and has an unknown biological function. Therefore, more efforts are required to understand FMDV RO formation, especially for FMDV 3A.

The FMD viral genome consists of a single open reading frame (ORF) that is translated into a polyprotein, which is then cleaved into structural proteins (VP0, VP3, and VP1) and nonstructural proteins (L ^pro^, 2A, 2B, 2C, 3A, 3B, 3C ^pro^, and 3D polymerase) [22]. Among these, 3A appears to be a peripheral membrane protein located on the ER with its N-terminus and C-terminus exposed to the cytoplasm [23]. The N-terminus contains two α-helices responsible for homodimerization and are essential for viral replication [24], while the C-terminus of 3A has been associated with host tropism [25,26]. However, the detailed mechanism and the biological function of 3A remain largely unknown. Currently, FMDV 3A is only known to be a multifunctional protein that participates in countering antiviral activity via the upregulation of LRRC25 and the autophagic degradation of RNA sensors [27]. In addition to LRRC25, G3BP1, and RNA sensors, the other 3A-associated interaction partners include DDX56, DCTN3 (dynactin subunit 3), vimentin, and RNA helicase A [28,29,30,31]. Despite intense study, many questions remain, such as the mechanism of LRRC25′s upregulation, the biological function of 3A–DCTN3 interactions, and the role of 3A in RO formation.

Protein trafficking from the ER to the ER-to-Golgi intermediate compartment (ERGIC) is mediated by the COPII pathway [32]. To generate COPII vesicles from the ER, Sar1 is recruited and activated by Sec12—a guanidine exchange factor—at the ER exit site (ERES). In other words, Sec12 exchanges the GDP of Sar1–GDP (inactive form) for GTP, inducing a conformational change in Sar1, with the N-terminus attaching to the ER membrane. Active Sar1 recruits the COPII inner coat proteins Sec23 and Sec24, as well as the outer coat proteins Sec13 and Sec31, to bend the membrane into vesicle-like structures [33]. Meanwhile, most cargo proteins and cargo receptors can interact with Sec24 to be loaded into the forming vesicles. To form a neck of vesicles, active Sar1 polymerizes. After Sar1 hydrolysis, vesicle scission is complete and the COPII vesicles are released from the ER toward the ERGIC [33]. The opposite direction of transport and the trafficking from the ERGIC to the Golgi are, instead, mediated by the COPI pathway. Therefore, the COPI/COPII machinery is crucial for protein secretion and vital for cells [32]. It was found that the machinery was also utilized by enterovirus and FMDV [8,16].

In our studies, the topology of 3A was first examined using homemade monoclonal antibodies (MAbs). Next, we proved that FMDV 3A expression alone could modify the ER into a specific vesicle-like structure. With the help of the APEX system, the detailed structure was examined under TEM. Moreover, without the N- or C-terminus of 3A, the ER remodeling was lost and the 3A was consequently retained at the ER. With the preservation of the regions near the hydrophobic region (HR), namely, aa 42–58 and aa 77–92, the truncated 3A was sufficient for reshaping the ER. The modification was associated with two unpublished host partners, Sar1 and Sec12, both of which are COPII factors. The depletion of Sar1 (or Sec12) and the expression of dominant-negative mutants hampered the remodeling. Furthermore, this is the first study to examine the features of 3A-induced vesicles from the web-like structure of the ER and to reveal the mechanism of their formation. These findings may serve to further the investigation of FMDV and other picornaviral ROs in the future.

## 2. Materials and Methods

### 2.1. Cells and Reagents

PK-15 cells and A549 cells were maintained in Dulbecco’s modified Eagle medium (DMEM) with 10% FBS. Rapamycin (Sigma, R0395) and digitonin (Sigma, D141) were purchased. The commercial antibodies used in this study included anti-FLAG mouse antibody (Sigma, F1804), anti-FLAG rabbit antibody (Sigma, F7425), anti-ß-actin (Cell Signaling Technology, #3700), anti-GFP (Invitrogen, MA5-15256), anti-calreticulin (LSBio, LS-B9387), anti-LC3B (Cell Signaling Technology, #2775), anti-Sec23A (Novus Biologicals, NBP2-34842), anti-Sec31A (Abcam, ab86600), anti-giantin (Sigma, SAB2100948), anti-Sec12 (LSBio, LS-C29886), and anti-Sar1 (Novus Biologicals, NBP2-20261).

### 2.2. Plasmids

#### 2.2.1. 3A-Associated Constructs

The primers used in this study can be found in Appendix A. To construct the expression plasmids pcDNA-3A(O99) and pcDNA-3A(O97), the 3A genes for O/TAW/97 (GenBank No. AY593835.1) and O/TAW/99 (GenBank No. AJ539136.1) were amplified (Phusion™ High-Fidelity DNA Polymerase, Thermo Scientific) from the O/TAW/97 cDNA pool and synthesized correspondingly. The 3A genes were subcloned into pcDNA-3.1(+) using the *Hind*III and *Xho*I sites with C-terminal His tag fusion; the endogenous *Nhe*I site within O/97 *3A* was further removed by site-direct mutagenesis without changing the amino acid sequence (GENEART site-directed mutagenesis system, Invitrogen, Carlsbad, CA, USA). The mGFP genes were inserted into pcDNA-3.1(+) with *Nhe*I/*Xho*I sites, in which a *Hind*III site was inserted between the mGFP gene and *Xho*I site, thereby generating pcDNA-mGFP. The 3A genes from pcDNA-3A(O99) and pcDNA-3A(O97) were individually subcloned into pcDNA-mGFP by *Hind*III/*Xho*I, thereby generating pcDNA-mGFP-3A(O99) and pcDNA-mGFP-3A(O97).

The d1D-2A-eGFP (d1A: deleted 1D gene; the last 33 nt of VP1 gene from O/TAW/97) cassette flanked by *Nhe*I-*Bam*HI and *Hind*III-*Xho*I was inserted into pcDNA-3.1(+) using the *Nhe*I and *XhoI* restriction enzymes, thereby generating the pcDNA-_2A_eGFP vector. Next, various truncations of 3A genes were inserted between *Hind*III/*Xho*I (or *Hind*III/*Xba*I) sites. For the 3A-eGFP_2A_ constructs, the genes and restriction sites from pcDNA-_2A_eGFP were rearranged. The d1D2A gene was subcloned into pcDNA-3.1(+) with *Nhe*I/*Xho*I sites, adding a *Bam*HI site to the 5′ end of the d1D2A gene. Next, the eGFP gene was inserted into the plasmid using *Nhe*I/*Bam*HI, and a *Hind*III site was added to the 5′ end of the eGFP gene. The variant truncated 3A genes were further inserted using *Nhe*I/*Hind*III.

The synthesized APEX2 gene, following the study by Lam et al. [34], was purchased and inserted into pcDNA-3.1(+) using *Nhe*I and *Xho*I. The indicated truncated or full-length 3A genes were inserted using *Xho*I and *Xba*I. For the GST fusion plasmids, the GFP genes in the variant mGFP- version and -eGFP_2A_ versions were replaced with GST genes using the *Nhe*I/*Hind*III and *Hind*III/*Xho*I enzymes, respectively.

#### 2.2.2. Host-Factor Plasmids

The Sar1a and Sec12 cDNA clones from swine and cattle were purchased from Origene (sSar1: NM_001031786; cSar1: XM_005226384; sSec12: XM_003125297; cSec12: NM_001076875). They were inserted into pcDNA-3.1(+) with a FLAG tag at the N-terminus. The sSar1 mutant constructs, including H79G, T39N, and QTTG (156-QTTG-159 to AAAA), were generated using the GENEART site-directed mutagenesis system (Invitrogen) with the indicated primers, while D198 was replaced with alanine using standard PCR with the primers BamHI-sSar1-F and XhoI-msSar1-D198A-R. The mutation of I41A for sSec12 was also performed with a GENEART site-directed mutagenesis system (Invitrogen).

#### 2.2.3. Others

The mCherry gene, flanked by the peptide sequences of calreticulin and the KDEL motif, was inserted into pcDNA-3.1(+) using *Nhe*I and *Hind*III to generate pcDNA-mCherry-ER. The plasmid for expressing mCherry-LC3B was a gift from David Rubinsztein (Addgene plasmid # 40827). The 2B and 2C genes were amplified from the O/TAW/97 cDNA pool and then subcloned into pcDNA-3.1(+) for pcDNA-2B and pcDNA-2C. These two plasmids were further inserted into mGFP genes using *Nhe*I and *Kpn*I, thereby generating pcDNA-mGFP-2B and pcDNA-mGFP-2C.

### 2.3. Monoclonal Antibody Preparation

BALB/c mice were immunized using purified SUMO-3ABC (O/97) and GST-3ABC (O/99), which were expressed from *E. coli*. Immunized spleen cells were fused with SP2/0-Ag14 myeloma cells. After 1–2 weeks, hybridoma supernatants were screened by IFA with 3A(O99), 3A(O97), and FMDV acetone-fixed cell plates. After at least two limited dilutions, the positive clones were further characterized. QA2, PA1, and T10E were selected for this study. Their isotypes all belonged to IgG2a/κ. The purified monoclonal antibodies were obtained according to the method used in a previous study [35]. The hybridoma was inoculated intraperitoneally into BALB/c for ascetic fluid and purified using Protein G (GE Healthcare). The purified antibodies were further conjugated with the fluorescent dye Dylight488 (Abcam, ab201799), according to the manufacturer’s instructions.

### 2.4. Western Blotting

Equal volumes of cell lysates or eluates from co-immunoprecipitation (co-IP) were mixed with a sample loading buffer (Bionovas, FA0020) and incubated at 95 °C for 5 min. The samples were separated by 13.5% SDS-PAGE, transferred to a nitrocellulose membrane (PALL, 79548), and blocked using 5% (*w/v*) skim milk in PBST at room temperature (RT) for 30 min. The proteins on the membranes were detected using the indicated antibodies, with overnight incubation at 4 °C, and the corresponding horseradish peroxidase (HRP)-conjugated secondary antibodies were used with incubation at RT for 1 h. Given that bands were masked by the light and the heavy chains of antibodies used in the co-IP step, secondary antibodies against native antibodies (Abcam, ab131366, which could not recognize mouse IgG1) or specific mouse native antibodies (Abcam, ab131368) were used for the elution samples.

### 2.5. Immunofluorescence Assay and Confocal Live-Cell Imaging

PK-15 or A549 cells, grown on a confocal µ-dish (ibidi, IB-81156), were transfected with the indicated plasmids for 24 h. For standard IFA protocols, cells were fixed using 4% paraformaldehyde in PBS for 15 min at 37 °C and further permeabilized using 0.5% TX-100 for 5 min at RT. For the examination of the 3A protein topology, fixed cells were permeabilized using 50 µg ml^−1^ of digitonin for 5 min at RT. After being washed with PBS three times, the samples were incubated with primary antibodies for 2 h at 37 °C, followed by incubation with appropriate secondary antibodies for 1 h at 37 ˚C. When double-labeling was performed, both antibodies were added together. The nucleus was stained with Hoechst stain (Invitrogen, H3569) at a 1:10,000 dilution for 15 min at 37 °C. All the imaging was performed under an Olympus IX-83 microscope connected to a CMOS (Complementary Metal Oxide Semiconductor) color camera. Live cells were maintained on the microscope stage at 37 °C.

### 2.6. Autophagy Induction

PK-15 cells in 24-well plates were transfected for 21 h. After washing with PBS, the medium was replaced with fresh DMEM, DMEM with 100 nM rapamycin, or starvation medium (20 mM HEPES, pH 7.4, 140 mM NaCl, 1 mM CaCl_2_, 1 mM MgCl_2_, 5 mM glucose, 1% BSA) for 3 h. The cells were collected in 50 µL of RIPA buffer (Bio Basic, RB4478); 5 µL of cell lysates were analyzed in a 6M urea 13.5% SDS gel, followed by Western blotting to detect LC3BI and LC3BII.

### 2.7. Transmission Electron Microscopy and Electron Tomography

PK-15 cells were seeded on an Aclar film (Electron Microscopy Science) in 6-well plates. After transfection for the indicated plasmids containing the APEX2 fusion protein gene for 24 h, the cells were rinsed using PBS, followed by 2% glutaraldehyde fixation in 0.1 M phosphate buffer (PB, pH 7.3) for 1 h on ice. After five washes with PBS, 20 mM glycine in PBS was added for 5 min to quench the free aldehyde groups. After another five washes, the cells were incubated with diaminobenzidine (DAB) solution with H_2_O_2_ (5 min for APEX2-3A, 15 min for APEX2-NHR, 30 min for APEX2-N2HRC, and 30 min for APEX2-N2HRC1). The cells were further rinsed in 0.1M PB (2 min × 5 times) and immersed in 2% osmium tetroxide in 0.1 M PB for 30 min. The cells were then rinsed in distilled water (2 min × 5 times) before overnight staining in a 2% aqueous solution of uranyl acetate. The cells in the monolayer were further subjected to a standard protocol of dehydration and embedding (Spurr’s medium) for TEM imaging. The specimen blocks were checked and trimmed under a stereomicroscope and ultra-thin sections (70 nm) were obtained from the area with positive signals (dark cells). The sections were observed using a transmission electron microscope (TEM, FEI Tecnai G2 TF20 Super TWIN) operating at 120 kV. For electron tomography, serial sections (200 nm) through the APEX2-positive cells were obtained [36]. Double-tilt electron tomography was performed with the TEM (FEI Tecnai G2 TF20), operating at 200 kV. The structures of the organelles were depicted using the Amira/Avizo software.

### 2.8. Recombinant Vaccinia Expression System

The protocol followed that of a previous study [37]. HTK^−^ cells were seeded in 6-well plates for 1 day, followed by vTF7-3 infection for the production of T7 polymerase within cells for 1 h. The indicated plasmids, containing a T7 promoter, were transfected to the cells by using TurboFect (Thermo Scientific, Carlsbad, CA, USA). After 20–24 h, the cells were ruptured by three freeze–thaw cycles in 500 µL of PBS with a protease inhibitor (Millipore, 539134). Finally, the cell debris was removed by 10,000× *g* centrifugation for 20 min.

### 2.9. Immunoprecipitation Assay

For each sample, 1 µg of anti-FLAG antibody (Sigma, F1804) or PA1 MAb was incubated with 10 µL of Protein G Mag Sepharose (GE Healthcare) in 500 µL of binding buffer (50 mM Tris, pH 7.5, 150 mM NaCl) with slow end-over-end mixing for 1 h at RT. Cell lysates that came from the vTF7-3 expression system or transfected PK-15 cells were applied for the co-immunoprecipitation assay. For the vTF7-3 expression system, a 200 µL sample was incubated with anti-FLAG antibody–magnetic beads overnight after the removal of the binding buffer. For the transfected PK-15 cells, which were harvested using 200 µL of RIPA buffer in each well of 6-well plates, 180 µL samples were applied. After mixing overnight at 4 °C, the magnetic beads were washed with PBS or RIPA three times. After replacement with a clean Eppendorf tube, the bead was washed with PBS again. Finally, the sample was eluted with 20 µL of 2% SDS and analyzed by Western blotting.

### 2.10. Knockdown Assay

A549 cells were seeded in a 6-well plate and then transfected with the indicated plasmids expressing shRNA, including pLAS2w.Ppuro (empty vector), Sar1-1 (target: CCAGTTCCTAGGACTCTACAA), Sar1-2 (target: CGTGAGATATTTGGGCTTTAT), Sec12-0 (target: GCTGGCCTAAAGATGCAATAA), and Sec12-4 (target: GTGTGCTTCAACCACGATAAT). The plasmids were purchased from the National RNAi Core Facility in Taiwan. After 24 h post-transfection, the cells were selected using 1 µg ml^−1^ puromycin for 3–4 days and maintained in DMEM with 0.5 µg ml^−1^ puromycin for more than 2 weeks. For siRNA transfection, cells were seeded in a 24-well plate 1 day before siRNA transfection, in accordance with the manufacturer’s instructions for Lipofectamine MessengerMAX Reagent (Thermo Scientific, LMRNA). The target sequence for non-target and Sar1 siRNA was as follows: non-target (UUCUCCGAACGUGUCACGU) and Sar1 (CCAGUUCCUAGGACUCUACAA), which were purchased from Biotools. To increase the knockdown efficiency, A549 cells were transfected repeatedly after 24 h of siRNA transfection. At 6 h after the last siRNA transfection, pcDNA-mGFP-N2HRC1 was transfected into the cells.

## 3. Results

### 3.1. Membrane Topology of Peripheral FMDV 3A Protein

Unlike the other picornaviruses, FMDV 3A is a peripheral membrane protein located on the ER [23,38]. However, it was useful to confirm this with another method. To probe further, we first produced three powerful anti-FMDV 3A MAbs, namely QA2, PA1, and T10E, against the O/TAW/97 (O/99) and O/TAW/97 (O/97) strains. The O/97 strain was a porcinophilic strain with ten amino acid deletions (aa 93–102) in the 3A protein, while the 3A of O/99 was relatively conserved with respect to another strain (Appendix A). Using variant monomeric GFP (mGFP) fused-3A truncated proteins (Figure 1a), the binding sites for these antibodies were determined by Western blotting (Figure 1b). QA2 MAb recognized the N-terminus of 3A (aa 1–41) for the O/99 and O/97 virus strains. PA1 and T10E could only interact with the C-terminus of 3A for O/99 and O/97 individually (Figure 1c), which is a highly diverse region between these two virus strains (Appendix A). Under digitonin treatment, the antibody could penetrate through the plasma membrane but not the ER or other organelles. For verification, calreticulin, a marker protein located within the ER lumen, was detected using a Triton X-100 treatment but not a digitonin treatment (Figure 1d). However, three anti-3A MAbs could detect 3A in all the conditions (Figure 1e), indicating that 3A was a peripheral membrane protein with both the N- and C-termini exposed to the cytoplasm.

### 3.2. FMDV 3A Protein Modified ER into Punctae

Based on co-localization testing, O’Donnell et al. had suggested that FMDV-induced 3A-containing vesicles may have originated from the ER [15], and Gonzalez-Magaldi et al. had further confirmed a connection between the FMDV 3A protein and the ER [23]. In our study, we found that the FMDV 3A immunofluorescence signal showed no co-localization with the ER marker protein calreticulin in 3A-transfected PK-15 cells (Appendix A), despite the 3A punctae displaying a web-like pattern. To identify the ER structure in live cells without alteration by fixation, we constructed mCherry-ER, in which mCherry was fused with the peptide signal of calreticulin and the KDEL motif (referring to the study by Roderick et al. [39]). A similar strategy was used in several previous studies [40,41,42]; afterwards, the cells were examined by immunofluorescence assays (IFAs) (Appendix A). After the co-expression of mCherry-ER and mGFP-3A, mGFP-3A punctae appeared at the site where the ER fragmented (Figure 2a). As shown in the right panel of Figure 2a, the ER structure was disrupted in the area (white arrow) displaced by mGFP-3A punctae, which were linked in a web-like pattern. In the early expression (3 h post-transfection), mGFP-3A showed a web-like structure, but it was only partially co-localized with mCherry-ER (Appendix A). Therefore, we considered that 3A punctae had been derived from the ER (Appendix A). Approximately 60% of the mGFP-3A-positive cells showed a punctate pattern, while the others appeared as a diffuse type (Appendix A). If the C-terminus of 3A had been truncated, they showed a clean reticular pattern and were highly co-localized with mCherry-ER (Figure 2b). These results support the concept that 3A molecules were located at the ER, and, by exerting an unknown function that required the C-terminus, 3A modified the ER into punctae. To determine the region essential for puncta formation, 3A was truncated into several parts. Considering that GFP interfered with the structure of the truncated 3A, the truncated 3A was fused to the C-terminus of mGFP (or _2A_eGFP) or the N-terminus of eGFP_2A_ (2A: FMDV 2A peptide, 19 aa) (Figure 2c and Appendix A). As a result, N2HRC1 spanning the region of aa 42–92 was sufficient for puncta formation (Figure 2c). However, it appeared as a web-like structure with a low expression level (approximately 28% of that for the transfected cells) (Figure 2d). This result suggested that the ability was dampened but not entirely abrogated by deleting the N1 and C2 regions.

In addition to 3A, the viral membrane proteins 2B and 2C were also examined. Consistent with previous studies [17,43], mGFP-2B would generally be located at the ER without obvious ER damage (Appendix A, left), as compared to mGFP-3A. mGFP-2C also showed co-localization with mCherry-ER and no ER fragmentation at a moderate expression level (Appendix A, right).

### 3.3. The 3A Punctae Were Distinct from Traditional COPII Vesicles or Autophagosomes

It has been reported that FMDV induced autophagy in favor of viral replication [44] and 3A expression upregulated LRRC25, an autophagy-related protein [27]. In addition, FMDV was postulated to initiate viral replication at the sites associated with the ER exit site (ERES), which was responsible for COPII vesicle formation [16]. Therefore, we aimed to exclude the possibility that 3A punctae were COPII vesicles or autophagosomes. According to IFA, 3A punctae from 3A (O99) were not co-localized with the COPII inner and outer coat proteins Sec23A and Sec31A (Figure 3a). Meanwhile, most cells for 3A expression led to Sec31A dispersion and downregulation (Figure 3a and Appendix A), which agreed with previous findings where FMDV infection had led to the dispersal and downregulation of Sec31 [16]. However, upon the activation of autophagy, LC3B-I was converted into LC3B-II and this became evident as punctae (Figure 3b,c; see starvation condition). Based on the LC3B-II/LC3B-I ratio detected by immunoblotting, the 3A protein mildly induced autophagy, as compared to empty vector transfection (Figure 3b,c). As they were not co-localized with mCherry-LC3B (Figure 3c), it was clear that mGFP-3A punctae were not autophagosomes. These results indicate that 3A punctae were specific structures and distinct from COPII vesicles and autophagosomes.

### 3.4. The Ultrastructure of the 3A Vesicle-like Structure as Observed Using Transmission Electron Microscopy

To gain better insight into the ultrastructure of the 3A punctae, we used electron microscopy and the APEX system, in which the APEX2 fusion protein could be directly stained using diaminobenzidine (DAB) directly. First, the patterns of APEX2-3A and APEX2-NHR were similar to those of the mGFP fusion constructs in IFA (Appendix A). The APEX2-NHR clearly illustrated the ER structure under electron tomography (Figure 4a, Appendix A). For the full length of 3A, as expected, the ER was disrupted into vesicle-like structures surrounded by APEX2-3A (Figure 4b; Appendix A). The diameter of these structures was heterologous and approximately 80.15 ± 26.54 nm (n = 83). Similarly, these modified structures were also found in cells expressing APEX2-N2HRC (Appendix A) and APEX2-N2HRC1 (Figure 4c). Moreover, in APEX2-N2HRC1-expressing cells, we also observed dark reaction products with segmented ER membranes (Figure 4c, white arrowheads), which were presumably in their state prior to reshaping. All the evidence pointed to 3A curving the ER membrane into vesicle-like structures and N2HRC1 preserving that ability.

### 3.5. COPII Factors, Sar1 and Sec12, Were Found to Be Novel 3A Interaction Partners

Given that polioviral and coxsackieviral 3A interacts with the COPI factor GBF1 [6,45] and that FMDV has been highly associated with the COPII pathway [16], we tested whether 3A would interact with the COPII factors Sar1 and Sec12. After the co-expression of 3A and Sar1 (or Sec12) in a recombinant vaccinia (vTF7-3) transient expression system, the interactions were examined by anti-FLAG co-immunoprecipitation (co-IP). As shown in Figure 5a,b, 3A from both the O/99 and O/97 virus strains was co-immunoprecipitated with recombinant Sar1 and Sec12 from swine and cattle. Therefore, these two novel interactions were probably independent of the host species and the virus strains. In addition, the reciprocal co-IP by PA1 MAb was tested (Figure 5c). Furthermore, in PK-15 cells co-expressing 3A with sSar1 (Sar1 from swine) or sSec12 (Sec12 from swine), co-localization was detected in double immunofluorescence using anti-FLAG rabbit antibody and QA2 MAb (Figure 5d).

Next, we mapped the binding sites on 3A for these two interactions. For Sec12, truncated 3A fragments with an N-terminal fusion of mGFP- or a C-terminal fusion of -eGFP_2A_ were co-expressed with sSec12 in PK-15 cells. After sSec12 was pulled down, mGFP-HR and C-eGFP_2A_ were detected, indicating that the binding site for Sec12 on 3A ranged from HR to the C-terminus (Figure 5e). Further truncation of the C-terminus narrowed this down to the C2 region, aa 93–153 (Figure 5e,g). For the Sar1–3A interaction, truncated GST-3A or 3A-GST was co-expressed with sSar1 in the vTF7-3 expression system due to its sufficient production level for sSar1. The results show that all three fragments, including the N-terminus, HR, and C-terminus, were associated with Sar1 (Figure 5f). More precisely, the region between the N1 and HR regions showed a high affinity for Sar1, while the N2 and C1 regions adjoining HR also interacted with Sar1 (Figure 5f,g).

### 3.6. Knockdown of Sar1 or Sec12 Inhibited Formation of 3A Punctae

Due to the strong association of Sar1 and Sec12, we reasoned that these two interactions could be responsible for the formation of 3A punctae. To verify our conclusion, a knockdown assay was performed. Considering the well-reviewed papers for knockdown in human species and the high abundance of the Sar1 protein in PK-15 cells, the human pulmonary cell line A549 was chosen for the knockdown assay. Furthermore, we chose mGFP-N2HRC1, as the weaker reshaping ability of N2HRC1 would be easier to manipulate and observe. It is well known that 3A is a multifunctional protein; the truncation of 3A and removal of its other abilities would help us to evaluate the impact on the depletion of Sar1 and Sec12 for puncta formation.

First, Sec12 and Sar1 were knocked down in A549 cells with shRNA or siRNA individually (Appendix A). As expected, puncta formation was inhibited in the Sec12 shRNA-A549 cell line (A0 cells) (Figure 6a). The re-expression of sSec12 restored the vesicular pattern, but not that of the Sec12 guanine nucleotide exchange factor-deficient mutant I41A (a mutation referred to previously [46]) (Figure 6a and Appendix A). These findings indicate that Sar1 activation may be essential for ER remodeling. Similarly, after Sar1 siRNA treatment, the number of punctae decreased in comparison to that after non-target siRNA treatment, while the re-expression of sSar1 could restore puncta formation (Figure 6b and Appendix A). It is well known that the COPII coat proteins Sec23 and Sec31 catalyze Sar1 GTPase activity, leading to the release of Sar1. Only when the Sar1 activation rate exceeds coat-induced hydrolysis does Sar1 polymerize to form a constricted neck, further exerting vesicle fission dependent on Sar1 hydrolysis. In short, the GTP cycle of Sar1 was essential for COPII vesicle formation [47]. Therefore, constitutive Sar1-GTP (H79G), constitutive Sar1-GDP (T39N), or defective organization Sar1 (156-QTTG-159 to AAAA) mutants could all block the COPII pathway. In our studies, during the co-expression of msSar1-T39N or msSar1-QTTG (m: mutated) with mGFP-N2HRC1, the formation of 3A punctae was significantly inhibited (Figure 6c). Nonetheless, msSar1-H79G did not disrupt puncta formation; instead, it enhanced it (Figure 6c). In addition, co-IP confirmed that all the mutated Sar1 showed a preserved interaction with 3A (Appendix A).

### 3.7. The Model of Active Sar1–3A Intercalating Complex for ER Remodeling

Based on our mapping analysis, we presumed that FMDV 3A might enhance Sar1 activation by tethering Sar1 to Sec12 with the N-terminus and C-terminus individually. After Sar1 activation, 3A intercalated with two active Sar1s to curve the ER membrane with the N2 and C1 regions (Figure 7). Therefore, mGFP-N2HRC1 preserved the ability for membrane curvature. In addition, 3A expression resulted in Golgi disassembly (Appendix A), as also reported by O’Donnell [15], since the blockade of the COPII pathway resulted in Golgi disruption due to failure in the transportation of PDIA3 [48]. In combination with the 3A-induced Sec31 dispersal (Appendix A), FMDV 3A should hijack Sar1 and disrupt the function of COPII machinery.

It has been reported that glycolipid glycosyltransferases (GGTs) directly interacted with Sar1 for transportation from the ER to the Golgi [49]. More precisely, the (R/K)X(R/K) motif at the GGT’s cytoplasmic tail bound to the pocket around D198, N126, and N94 at Sar1. The D198A mutation of Sar1 abrogated the interaction for GGTs, resulting in the accumulation of GGTs at the ER [49,50]. However, the D198A mutation should not affect most of the protein trafficking from the ER to the Golgi. Surprisingly, we found that msSar1-D198A led to the inhibition of puncta formation (Figure 8a). Most importantly, the D198A mutation significantly decreased the ability to bind to the N2 and C1 domains of 3A (Figure 8b). The D198 residue, the last amino acid of Sar1, was positioned near the membrane, which fit our model.

## 4. Discussion

Several advanced studies have revealed that viruses can modify host organelles to serve as replication sites or provide viral RNP transport [1,51,52,53,54]. How viruses modify a host endomembrane has been a focus of research in the field, especially in the case of positive single-stranded RNA viruses. Notable studies on enteroviral replication organelles (ROs) have been conducted. Although some components, such as ACBD3, PI4K, OSBP, and cholesterol, have proved essential for enteroviral ROs, the actual causes of RO formation are not yet fully understood [3,13,55]. Currently, the viral 3A protein is thought to be a key component and one of the initiators. However, FMDV 3A is unique compared to proteins from other picornaviruses [23].

In a study by Gonzalez-Magaldi et al., they utilized a co-localization test, a fluorescence recovery after photobleaching (FRAP) analysis, a fluorescence loss in photobleaching (FLIP) analysis, and a membrane-bounded fraction analysis to support the idea that 3A was a peripheral ER membrane protein [23]. In this study, we have demonstrated new evidence showing that FMDV 3A is a peripheral membrane protein by using digitonin and monoclonal antibodies. Moreover, FMDV 3A modified the ER into vesicle-like structures, as determined by fluorescent live microscopy and electron tomography with the APEX system. Following up on a study reporting that FMDV was associated with the COPII pathway [16], we found that two COPII factors, Sar1 and Sec12, interacted with FMDV 3A based on a co-IP and co-localization test, and these two interactions were associated with 3A-induced ER remodeling, as shown by knockdown and re-expression assays. Finally, we built a model to explain how FMDV 3A modified the ER.

In our hypothesized model, 3A tethered Sar1 to Sec12 for activation via the N-terminus and C2 region. Due to the failure of the active Sar1 pull-down assay, this assumption should be validated by future research. However, Sar1 activation is necessary for remodeling, as functionally deficient Sec12 (I41A-msSec12) would be unable to restore 3A puncta formation in Sec12-depleted cells. After attaching to the ER membrane, two active Sar1s interacted with the 3A protein in the N2 and C1 regions, causing membrane curvature—instead of recruiting Sec23 and Sec24. Therefore, mGFP-N2HRC1 preserved the ability for ER remodeling. Constitutively active Sar1 (msSar1-H79G) enhanced 3A puncta formation, while constitutively inactive (T39N) and polymerization-deficient (156-QTTG-159 to AAAA) Sar1 inhibited that formation. In addition, 3A expression disrupted the COPII pathway, inducing Sec31 dispersal and Golgi disorganization. Most importantly, the D198A mutation of Sar1 inhibited the interaction with the N2 and C1 regions of 3A, thereby inhibiting puncta formation. The D198A mutation did not inhibit Sar1 activation or most protein trafficking that is dependent on COPII vesicles [50]. As the position of D198 for active Sar1 is located near the membrane, the amino acid fits the positions of the N2 and C1 regions of 3A that adjoin the hydrophobic region. Although this study was not focused on examining the specific membrane-remodeling processes, future experiments using cell-free techniques [46,56] and giant unilamellar vesicles (GUVs) are warranted to highlight our proposed model [57].

The results of the knockdown assay also indicated that 3A puncta formation required Sar1 and Sec12, although we had numerous difficulties. For example, an attempt to use PK-15 cells in an shRNA strategy was unsuccessful, possibly due to the narrow window between lethal doses of puromycin and the efficient knockdown condition for Sar1 and Sec12. Moreover, we were unable to find available antibodies against endogenous Sec12 from PK-15 cells in Western blotting. Therefore, A549 cells were selected for this experiment. In addition, the shRNA Sar1 A549 cell lines did not survive the puromycin selection; therefore, the siRNA strategy was adopted as an alternative for the Sar1 knockdown assay.

Due to the biosecurity issues in Taiwan, our studies were restricted by virus-free regulations, so our hypothesis could be in question. However, a study by Midgley et al. presented reliable experiments for FMDV infection [16]. They showed that the knockdown of Sar1 significantly inhibited FMDV infection in IBRS2 cells. The T39N mutation of Sar1 inhibited FMDV infection in IBRS2 cells, while the H79G mutation did not. FMDV infection led to the dispersal of outer coat protein Sec31 labeling in HeLa cells. These results agree with our model’s prediction that FMDV 3A utilized and required active Sar1 for ER remodeling, resulting in interference with the COPII pathway. The ER remodeling should be the initiation of RO formation, which is essential for viral replication.

The ER is commonly utilized and modified by viruses such as hepatitis C virus, influenza virus, Zika virus, and reovirus [40,51,58,59]. Zika virus provides a simple but straightforward example of how a virus modifies the ER. Zika NS1 proteins, located within the ER lumen, homodimerize to induce ER invagination, which is probably similar in dengue virus [59]. However, most other viruses may be more complicated. For example, reovirus-induced ER modification has required both σNS and μNS proteins [40]. For poliovirus (an enterovirus), the formation of the modified structure mimicking the RO required the co-expression of 3A and 2BC [60]. We also considered that the mechanism of RO formation could have been associated with 2B and 2C and required further exploration.

Overall, to our knowledge, this study is the first to demonstrate that the FMDV 3A protein modified the ER into vesicle-like structures. The modification was synchronized by two unrecognized interactions, 3A–Sec12 and 3A–Sar1. In combination with other indirect evidence, we presumed that 3A enhanced Sar1 activation and further interacted with two active Sar1 proteins such that the membrane was bent into vesicle-like structures, which may be the first and most important step in RO formation during FMDV infection.

## Figures and Tables

**Figure 1 viruses-14-00839-f001:**
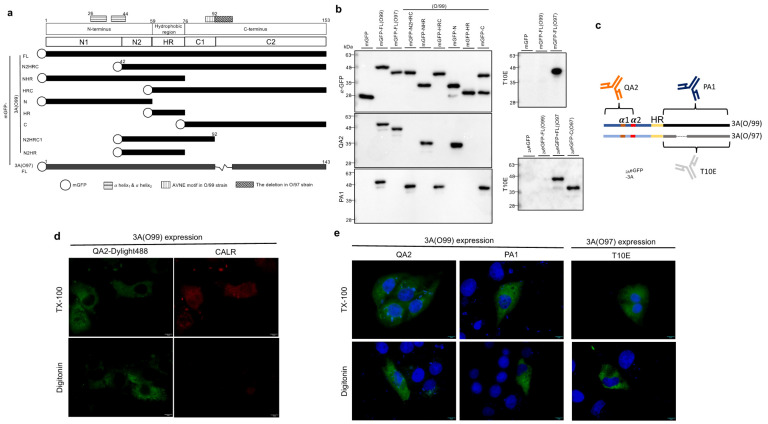
(**a**) Diagram representing the different truncations of O/99 3A or O/97 3A fused with monomeric GFP (mGFP). We set the alpha-helix2 region and the deletion region in O/97 as the turning points to separate the N- and C-termini into N1/N2 and C1/C2. All the truncation forms here used the O/99 normal virus strain as a template, including the aa 1–41 deletion (mGFP-N2HRC), aa 1–41 and aa 93–153 deletion (mGFP-N2HRC1), N-terminus deletion (mGFP-HRC), C-terminus deletion (mGFP-NHR), hydrophobic region (mGFP-HR), C-terminus (mGFP-C), and N-terminus (mGFP-N). (**b**) The constructs, expressed in PK-15 cells, were used to examine homemade anti-3A MAbs by Western blotting. QA2 could not recognize N2HRC, indicating the binding site located at the region within aa 1–41 of 3A, while PA1 recognized the C-terminus of 3A from O/99. The binding site for T10E should be the C-terminus of O/97 3A. (2AeGFP: eGFP fused with FMDV 2A peptide in N-terminus.) (**c**) The schematic of the MAb binding sites on 3A protein. (**d**) PK15 cells were transfected for expressing 3A from O/99 or O/97. After fixation, the cells were permeabilized using Triton X-100 or digitonin. Anti-calreticulin rabbit antibodies were used to stain the cells in the control group, followed by anti-rabbit secondary antibodies (red) and MAb QA2 conjugated to Dylight 488 (green) for dual labelling. (**e**) For the examination of the topology of 3A, the cells were stained with the indicated MAb (QA2, PA1, or T10E), followed by anti-mouse secondary antibodies. The nucleus was stained using Hoechst stain.

**Figure 2 viruses-14-00839-f002:**
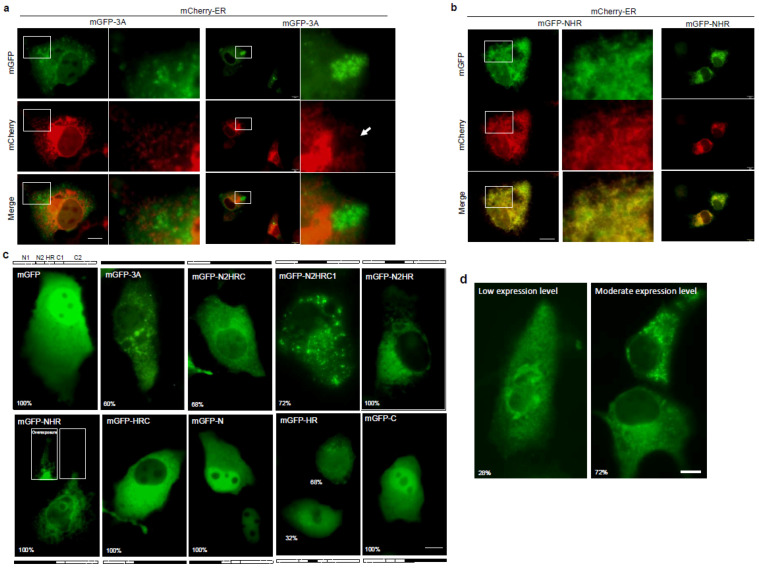
(**a**,**b**) Images obtained in PK-15 live with mCherry-ER and mGFP-3A (or mGFP-NHR) co-expressed. The white arrow indicates the area of ER fragmentation. (**c**) Images of the variant truncation of 3A, in which the N-terminus was fused to mGFP. The mGFP diffused within the cytoplasm and nucleus. More than half of the cells expressing mGFP-3A, mGFP-N2HRC, and mGFP-N2HRC1 (60%, 68%, and 72%, respectively) showed multiple punctae in the population. In addition, mGFP-N2HR and mGFP-NHR showed reticular patterns, while only 68% of the mGFP-HR-expressing cells were reticular and the others appeared to be diffuse-type. The mGFP-HRC protein distributed diffusely in the cytoplasm, and mGFP-N and mGFP-C in the cytoplasm and nucleus. (**d**) N2HRC1 expression in PK-15 cells showing reticular and punctae patterns at low (with longer exposure time) and moderate expression levels. Scale bar, 10 μm.

**Figure 3 viruses-14-00839-f003:**
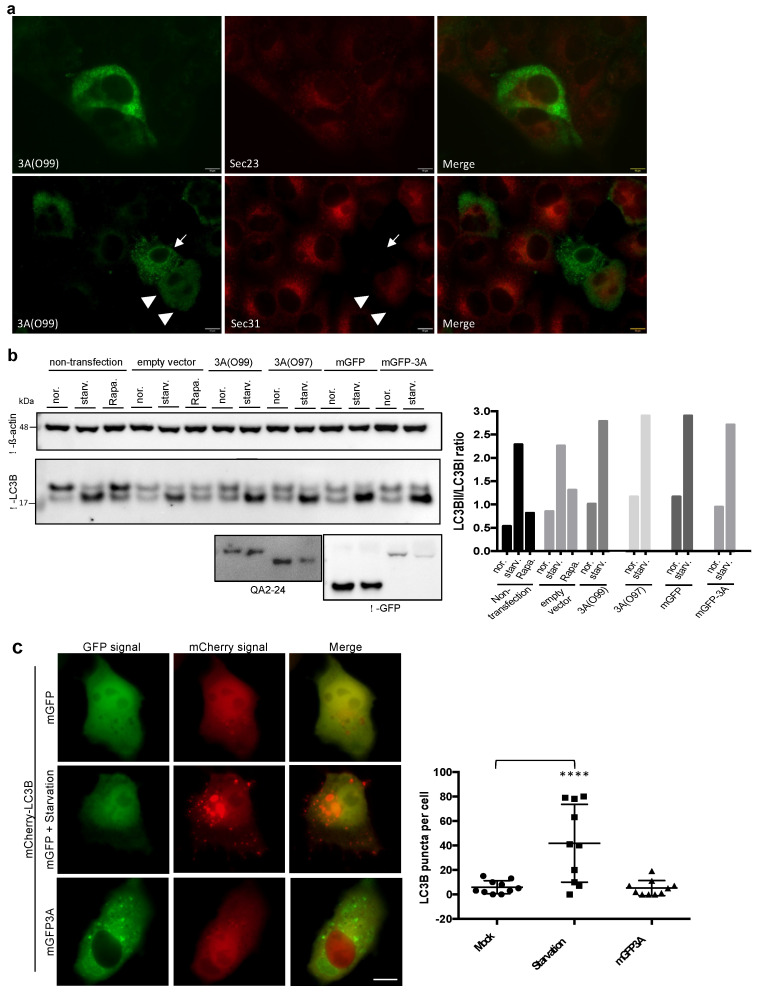
(**a**) PK-15 cells expressing 3A (O99) were stained with specific anti-Sec23A or Sec31A rabbit antibodies, followed by anti-rabbit antibodies conjugated with Alexa Flour 594. Finally, the 3A (O99) protein was further identified using QA2-Dylight488. The white arrow indicates the cell expressing 3A showing downregulation of Sec31, while white arrowheads indicate those showing Sec31 dispersion. (**b**) PK-15 cells were transfected using an empty vector (pcDNA-3.1(+)), pcDNA-mGFP, pcDNA-3A, or pcDNA-mGFP-3A for 21 h. The cells were further incubated in either normal, 100 nM rapamycin, or starvation condition medium for 3 h. The LC3B proteins from these cell lysates were analyzed by Western blotting and quantified using the ImageJ software. (**c**) PK-15 cells co-expressing mCherry-LC3B with mGFP or mGFP-3A. The cells expressing mGFP in starvation condition medium for 3 h were regarded as a positive control for autophagy activation and showed much more LC3B punctae than did the non-treated cells, but there was no significant increase in mGFP-3A overexpression. Scale bar, 10 μm. **** *p* < 0.0001.

**Figure 4 viruses-14-00839-f004:**
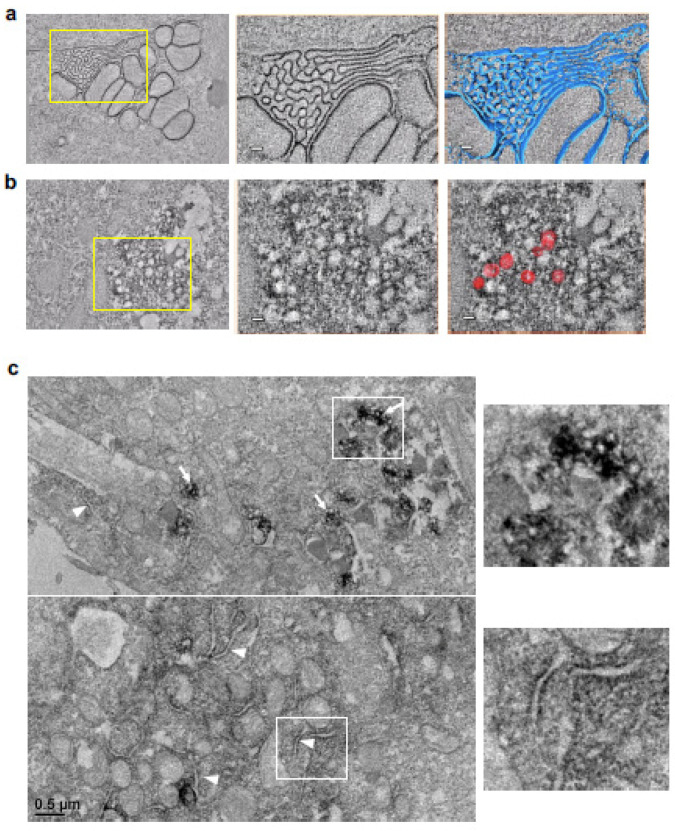
Electron tomography images of (**a**) APEX2-NHR and (**b**) APEX2-3A. The serial section was obtained from PK-15 cells expressing APEX2-NHR or APEX2-3A (full-length of 3A). The ER structures with DAB reaction products for APEX2-NHR (blue) were illustrated in the Amira/Avizo software, while eight vesicle-like structures with APEX2-3A were manually illustrated (red) as representative examples. (**c**) TEM images of APEX2-N2HRC1. White arrows indicate vesicle-like structures with APEX2-N2HRC1. In addition, segmented ER was found and marked by a DAB reaction (white arrowheads).

**Figure 5 viruses-14-00839-f005:**
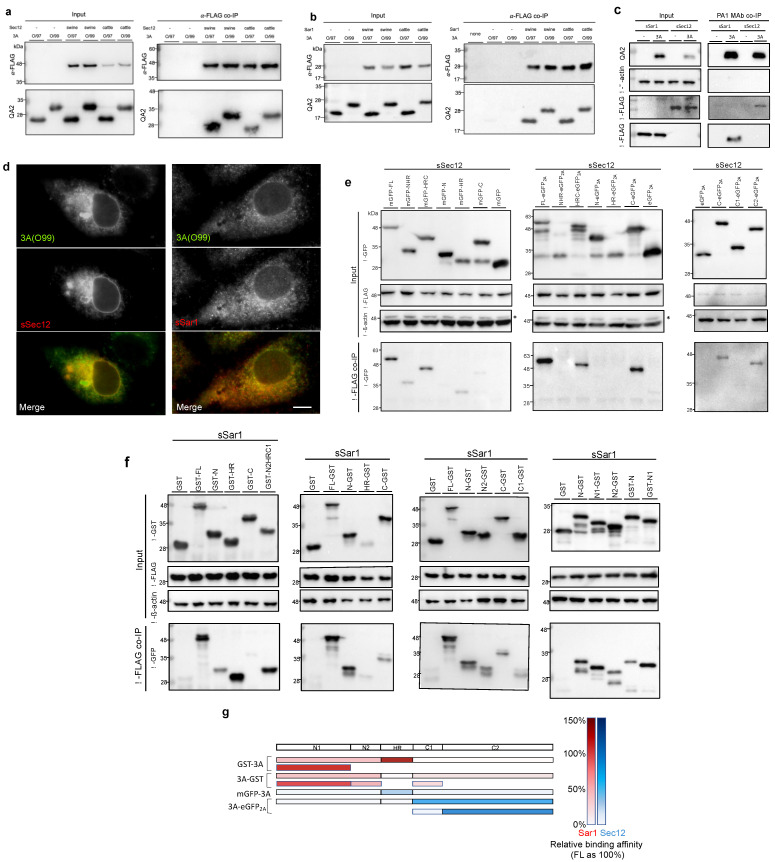
The examination of interactions between 3A and Sec12 or Sar1. (**a**) In the vTF7-3 expression system, HTK cell lysates for the co-expression of 3A from O/97 or O/99 and Sec12 from swine or cattle were applied to anti-FLAG co-immunoprecipitation assays: empty vector, pcDNA 3.1(+), none: non-transfection. (**b**) The examination for Sar1 was performed using the same procedure. (**c**) Reciprocal co-immunoprecipitation was performed with anti-3A MAb PA1. (**d**) PK-15 cells co-expressing 3A (O99) with sSec12 or sSar1. The cells were double-labeled for 3A and recombinant COPII factors using MAb QA2 and anti-FLAG rabbit antibodies, followed by anti-mouse and anti-rabbit secondary antibodies conjugated with Alexa Fluor 647 (green) and Alexa Fluor 594 (red), correspondingly. Scale bar, 10 μm. (**e**) PK-15 cells co-expressing sSec12 with variant truncation GFP version for anti-FLAG co-immunoprecipitation. *: residual signal for Sec12. (**f**) Co-expression of sSar1 with the truncated GST variants in the vTF7-3 system was applied for co-IP. (**g**) The diagram represents the relative binding affinity of each part of Sar1 or Sec12. The protein level of full-length 3A from the elution was regarded as 100%. Due to a relatively low expression level, HR-GST for Sar1 was excluded.

**Figure 6 viruses-14-00839-f006:**
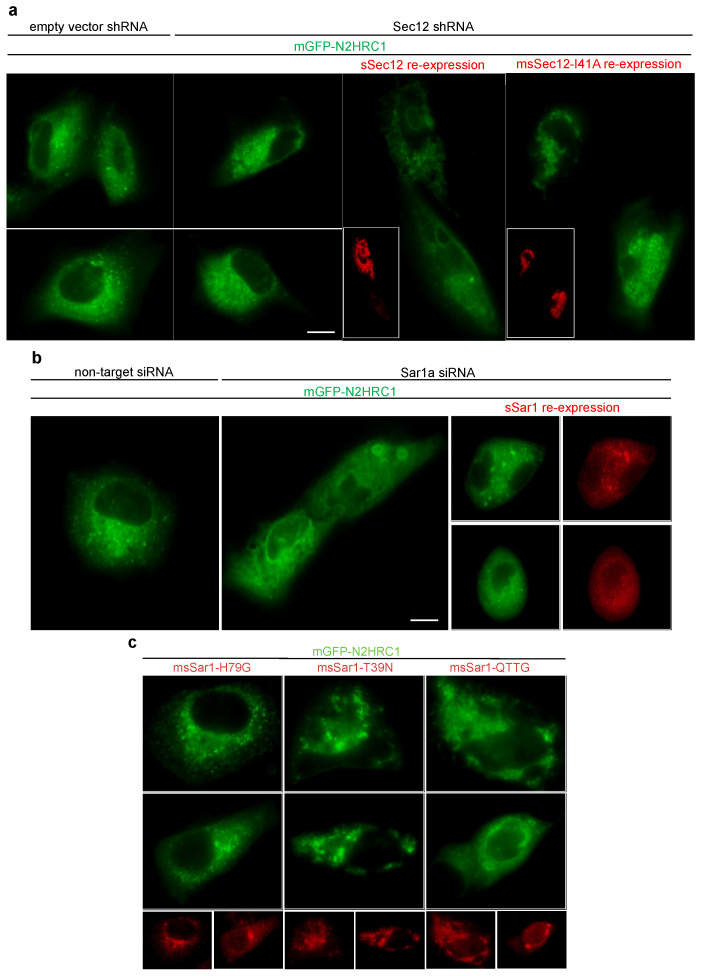
The fluorescence patterns of mGFP-N2HRC1 in A549 cells were examined under different conditions. (**a**) mGFP-N2HRC1 was expressed in A549 cells, constitutively expressing control shRNA or Sec12 shRNA. It showed a reticular pattern with punctae for pLAS2w.Ppuro (empty vector) cell lines, while the punctae significantly decreased in the Sec12 shRNA cell line (A0 cells). For the re-expression assay, the A0 cells were co-expressed with mGFP-N2HRC1 and sSec12 (wild type or I41A). The sSec12 proteins were identified using an anti-FLAG mouse antibody (red). (**b**) A549 cells were treated with siRNA (non-target; targeting Sar1), followed by mGFP-N2HRC1 expression. For the re-expression assay, sSar1 and mGFP-N2HRC1 were co-expressed after Sar1 knockdown and IFA identified recombinant Sar1, as above. Scale bar, 10 μm. (**c**) A549 cells were co-transfected for mGFP-N2HRC1 and dominant-negative msSar1 (mutated sSar1), which was labeled with anti-FLAG antibodies (red).

**Figure 7 viruses-14-00839-f007:**
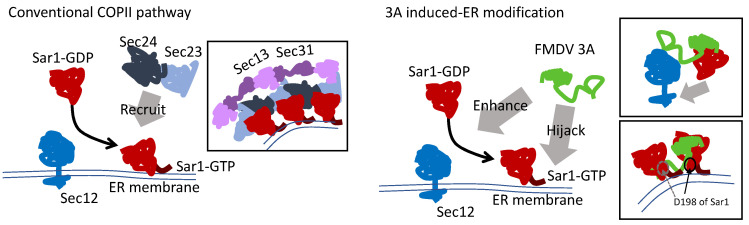
In the conventional COPII pathway, after activation by Sec12, Sar1 attaches to the ER membrane with the N-terminus, further recruiting the inner coat proteins Sec23 and Sec24. Sec24 recruits the COPII outer coat proteins Sec31 and Sec13 to bend the membrane into a vesicle-like structure. In FMDV 3A-expressing cells, we hypothesized that FMDV 3A modifies the ER into a vesicle-like structure, including Sar1 activation/enhancement and 3A—a Sar1–GTP intercalating complex. That is, 3A enhances Sar1 activation. Next, 3A intercalates into two active Sar1s to bend the membrane into vesicle-like structures.

**Figure 8 viruses-14-00839-f008:**
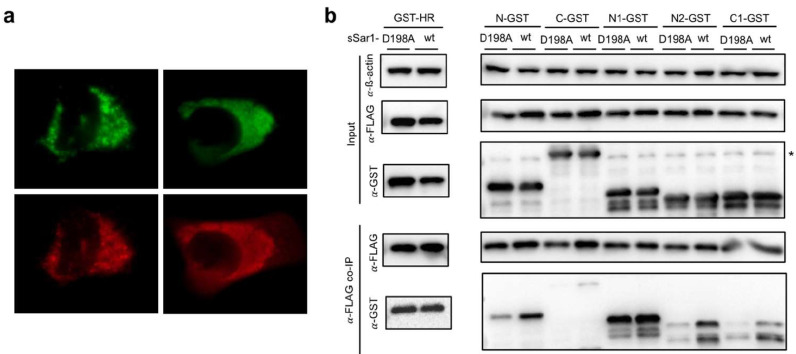
(**a**) msSar1-D198A (red) and mGFP-N2HRC1 (green) were co-expressed in A549 cells, followed by IFA for identifying recombinant Sar1 with anti-FLAG antibody. (**b**) The variant of truncated 3A fused with GST was co-expressed with wild-type sSar1 or the D198A mutation of Sar1. After anti-FLAG co-IP, fragments of 3A were examined by anti-GST antibodies. *: the nonspecific band.

## Data Availability

The data presented in this study are available on request from the corresponding author.

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
