# Peer review of "Foot-and-Mouth Disease Virus 3A Hijacks Sar1 and Sec12 for ER Remodeling in a COPII-Independent Manner"

_viruses, 2022, doi:10.3390/v14040839_

Round 1

Reviewer 1 Report

The manuscript submitted by Lee et al. discusses the role of FMDV 3A in replication organelle formation. The authors show that expression of 3A results in the rearrangement of the ER and was dependent on the presence of the C-terminus. Additionally, the authors show that remodelling of the ER was dependent on 3A interactions with host proteins, Sar1 and Sec12. Overall, the authors present an interesting study with compelling data. However, there are several items that should be addressed prior to publication.

Major:

The authors state that 3A exists as a peripheral membrane protein. However, the staining in figure 1 appears diffuse. The authors should perform western blots on fractionated cell lysates expressing their truncated constructs to verify localization of the different 3A proteins.

The authors state that 3A is responsible for curving the membrane. However, there is no direct evidence for this. In order to make this claim, the authors should mix liposomes with purified 3A to determine if 3A can directly impact curvature of membranes.

Authors state that knockdown of Sar1 demonstrates a decrease in 3A-dependent punctae formation. Only representative images are shown for a few cells; therefore, the results should be quantified to represent more than a single field of view.

To claim that Sar1 and Sec12 are required for replication organelle formation, the authors should perform infection experiments in cell depleted of these host factors, including quantification of the punctae formed under the conditions where mutant sSar1 is expressed to solidify these results.

The abstract overstates the findings by claiming that “two active Sar1 were connected by 3A with regions of aa 42-59 and aa 76-92, causing membrane curvature. There is no direct evidence supporting this major claim.

Minor:

The manuscript would benefit from being assessed for English and grammar.

Where was APEX2 plasmid obtained from?

What concentration of Hoechst was used for nuclei staining?

Is it accurate to state the ER is collapsed?

Fig 3b: There is a horizontal black bar in the top blot between lanes 1 and 2.

The left images of Fig 5d are significantly overexposed. The authors should provide images with more moderate exposure.

The authors should explain why they use a vaccinia virus system rather than the constitutively active CMV promoter plasmids used in previous studies

Reviewer 2 Report

In the manuscript by Lee et al,  the authors describe a novel mechanism in which overexpressed FMDV 3A modifies the endoplasmic reticulum to induce the formation of vesicle-like structures. The authors support the findings experimentally by using various biochemical assays and microscopy work. The work presented in this study offers relevant information toward the understanding of replicative organelles during FMDV infection, however the study lacks functional assays to highlight the relevance of this study. Although the authors indicate that current lab regulations do not allow the work with select agents such as FMDV, other assays focused on protein secretion could have added more physiological relevance to this study. The authors provided a very thorough analysis on the molecular mechanisms that are necessary for the ER remodeling thru FMDV 3A. The writing of the manuscript is very sloppy, lacks flow and makes the reading extremely difficult. Some of the supplementary figures should be included as panels in the figures of the manuscript to add more clarity.  The manuscript has a lot of figures and panels supporting the conclusions stated in the manuscript, but the writing is not concise and offers more confusion than clarity. Please see my comments below:

Given that the manuscript focuses on the ER-Golgi traffic, the author should provide a very simple introduction on this cellular process. There are a lot of acronyms without definitions.

“Although the mechanism of RO formation differs in different virus genera…enterovirus could represent a paradigm”. This sentence is badly written. There are too many examples of bad sentence construction. The authors should revise and edit the manuscript further. If the study on enteroviruses “could represent a paradigm” why do you study this topic? This is vey confusing.

The paragraph before materials and methods state: “All in all, this is the first study to examine the features of 3A induced structures and the mechanism of their formation”.. the authors should define what these structures are. The reader does not have a clue on the structures that are induced by 3A. I recommend the authors to fully describe the normal structures of the ER and then highlight the changes that are induced by FMDV 3A overexpression. This will add more clarity to the manuscript.

In the immunofluorescence microscopy work there are some inconsistencies, some figures show the stain of the nucleus and in others they are not shown. I wish they kept the Hoescht/DAPI stain for all of the pictures shown by confocal/immunofluorescence microscopy. This is especially important for some images (Figure 6) where there is an indication that cell viability maybe impacted. In others there are some nuclear structures that are being displayed in the green signal. Also, in Figure 2C, nuclear detection in the green channel is detected but not mentioned in the manuscript.

The authors mention that in most cells 3A expression would lead to Sec31A dispersion and downregulation but there is no indication in Figure 3a of these changes. The authors should provide some arrows to clearly indicate these changes.

In Figure 5 C, why Flag detection was done in two separate blots. Shouldn’t the signals be detected in one blot. This is confusing.

In Figures 7 the authors provide some conclusions where they indicate that 3A intercalates into 2 Sar1 to bend the membrane into a vesicle like structure. Experiments to suggest bending of the ER membrane were not conducted in this study.

Round 2

Reviewer 1 Report

Revised manuscript is acceptable.

Author Response

We have consigned English-editing suggested by MDPI, and please see the new revised file. Thank you.

Reviewer 2 Report

The authors have now presented a more polished version of the manuscript. They have answered most of my suggestions during the first round of this review process. However, there are still minor deficiencies that are related with the English style. Once again, the manuscript is very complex and attention to detail is a must for the reader to find more clarity on the results that are being described. I would ask the authors to do another round of editing and perhaps ask a English native speaker to read the manuscript one more time. Below are some examples of sentences that are badly written and can confuse the reader:

"However, Sar1 activation is necessary for remodeling, for functionally deficient Sec12 (I41A-msSec12) would be unable to restore 3A punctae formation in Sec12 depleted cells. After attaching to the ER membrane, two active Sar1 interact with 3A protein in the N2 and C1 regions, causing membrane curva[1]ture, instead of recruiting Sec23 and Sec24..." THIS PARAGRAPH IS VERY CONFUSING

“Honestly, there is no direct evidence to prove the model in our study...” THIS PARAGRAPH IS VERY CONFUSING.  Consider changing to: Although this study is not focus on examining specific membrane remodeling processes, future experiments using cell-free techniques (references) and giant unilamellar vesicles (GUVs) are warranted to highlight our proposed model. 

In the materials and methods section, the authors do not indicate either the use of A549 cells nor the media conditions to grow these cells. 
